# Determinants for Deployment of Climate-Smart Integrated Pest Management Practices: A Meta-Analysis Approach

Haruna Sekabira [1,*], Ghislain T. Tepa-Yotto [2,3], Rousseau Djouaka [2], Victor Clottey [4], Christopher Gaitu [5], Manuele Tamò [2], Yusuf Kaweesa [6] and Stanley Peter Ddungu [6]

1    International Institute of Tropical Agriculture (IITA-Uganda), Plot 15, East Naguru Road, Kampala P.O. Box 7878, Uganda
2    Biorisk Management Facility (BIMAF), International Institute of Tropical Agriculture (IITA-Benin), Cotonou 08-01000, Benin; g.tepa-yotto@cgiar.org (G.T.T.-Y.); r.djouaka@cgiar.org (R.D.); m.tamo@cgiar.org (M.T.)
3    Ecole de Gestion et de Production Végétale et Semencière (EGPVS), Université Nationale d'Agriculture (UNA), Kétou 43, Benin
4    CABI, Cantonments, Accra P.O. Box CT 8630, Ghana; v.clottey@cabi.org
5    PPRSD (PPRSD), Ministry of Food and Agriculture (MoFA), Accra P.O. Box M37, Ghana; chrisanthus88@yahoo.com
6    LADS Agricultural Research Consult, Kampala P.O. Box 4235, Uganda; yusuf.kaweesa@gmail.com (Y.K.); ddungustanley@gmail.com (S.P.D.)
*    Correspondence: h.sekabira@cgiar.org

**Abstract:** Following the development and dissemination of new climate-smart agricultural technologies to farmers globally, there has been an increase in the number of socio-economic studies on the adoption of climate-smart integrated pests' management (CS-IPM) technologies over the years. In this study, we review empirical evidence on adoption determinants of CS-IPM technologies and identify possible science–policy interfaces. Generally, our review shows that socioeconomic and institutional factors are influential in shaping CS-IPM adoption decisions of farmers. More specifically, income was found to positively influence the adoption of CS-IPM technologies while land size owned influences CS-IPM adoption negatively. Registered land tenure (registered secure rights) positively influences CS-IPM technologies' adoption, implying that efficient land markets enable competitive and fair distribution and access to land, more so by the vulnerable but efficient smallholder producers that do indeed increase the adoption of CS-IPMs technologies. Social capital, achieved via farmers' organizations is also central in fostering CS-IPM technologies' adoption, just as market reforms that minimize market failures regarding access to credit, labor, and agricultural information, which could indirectly hinder farmers' use of CS-IPM practices. Functional extension systems that improve farmers' awareness of CS-IPM do also improve CS-IPM technologies' adoption. However, the adoption of CS-IPM technologies in Ghana and Benin is slow-paced because of factors like lack of access to farm inputs that facilitate uptake of these technologies, lack of credit facilities, and limited extension services, among others. Interestingly, our review confirms that CS-IPM technologies do indeed reduce and minimize the intensity of pesticide usage and foster ecosystem (environmental and human) health. Therefore, this review unearths strategic determinants of CS-IPM adoption and makes fundamental guidance around climate-smart innovations transfer and environmental policies that should be prioritized to curb environmental pollution and ensure agricultural ecosystems' sustainability.

**Keywords:** socio-economic determinants; agricultural technologies; climate-smart; integrated pest management technologies (CS-IPM)

## 1. Introduction

Many vulnerable populations around the globe depend on agriculture for subsistence [1]. Although improvements in agricultural productivity in recent decades have

enabled considerable reductions in global hunger [2], the agricultural sector still faces many challenges, including trade barriers, scarcity of resources, breakdown of market systems, unstable and ineffective socio-economic policies, insecurity, increasing population pressure, unsustainable and soil degrading agronomic practices, the dilapidation of the environment, weather unpredictability, and climate change [3]. These barriers to agricultural production risk food security and safety of rural populations, especially of the resource-poor smallholder farmers [4]. The adoption of climate-smart practices might help mitigate these risks [3,5]. Climate-smart practices are often combined with Integrated Pest Management (IPM) practices that foster the rational and parsimonious use of pesticides, thus reducing dependence on conventional chemical pesticides [4]. Climate-Smart Agriculture (CSA) is addressing these challenges by helping the agricultural sector to adapt to climate change and reduce emissions, concepts that are receiving wider global endorsement and are progressively gaining in importance, especially in strategic farming systems such as the cocoa production system in Ghana [1,3,6]. CSA aims to optimize social, environmental, and economic benefits while sustainably bringing development by anchoring on three key strategic pillars: (1) sustainably improve agricultural productivity and household income; (2) adjust to and build formidable household resilience to climate change; and (3) minimize and/or eliminate greenhouse gases emissions if compared to conventional agricultural practices. Relying only on chemical pesticides for crop protection against pests to guarantee improvements in agriculture leads to pesticide overuse, thus leading to chronic environmental poisoning and deterioration [7]. Hence, effective climate-smart agriculture must be climate-smart integrated pest management (CS-IPM) compliant. The most important aspect of any CS-IPM approach is that the CS-IPM practice must integrate the context of the affected ecosystem into the pest control strategy, with a major goal being to use naturally available resources sustainably and to control pests while avoiding further outbreaks [8]. Furthermore, an effective CS-IPM practice must integrate spatial, temporal, and environmental dimensions of the target context. Hence, CS-IPM practices must be explicit concerning the target crop, the region where the crop is grown, and the time during the crop's growth when the practices must be applied. Based on such a ground, crop rotation is one of the noble examples of CS-IPM practices. Unlike other pest control measures like Chemical Crop Protection, which can be classified as regular IPM, CS-IPM hinges on nature to achieve effective pest control [8]. CS-IPM practices bring long-term stability between the environment and agricultural production systems, leading to sustainable food systems [4,8,9]. Since most of the vulnerable populations, especially in the global south, are dependent on agriculture, which is heavily dependent on the natural environment, it is prudent that innovations maximize the well-being of the environment equally as economic gains to achieve sustainable food systems [4]. Henceforth, CS-IPM knowledge must be developed, and appropriately packaged for adoption by farming communities and policy. Sadly, much of the developing world has been slow at adopting CS-IPM innovations and practices due to the huge knowledge gaps that exist [1,3,4,9]. In this meta-analysis review, we contribute to closing this knowledge gap by understanding the key socio-economic factors that influence the deployment of CS-IPM practices. We use the internet as the primary source of articles reviewed for this study. However, CS-IPM being a recent innovation in agriculture, articles on CS-IPM were limitedly available, especially in Africa. Although our focus was global, we had a keen consideration for Ghana and Benin, because these are action countries selected for the implementation of the "Accelerating the Impact of CGIAR Climate Research for Africa" (AICCRA) project, which is a new CGIAR (Consultative Group on International Agricultural Research) initiative aiming to increase access to climate information services and authenticated climate-smart agriculture practices in Africa. This study is therefore central in advising priorities and investment decisions of AICCRA and similar projects, as well as strategic policies intended to preserve the environment and sustainable food systems globally through CS-IPM practices.

*Brief Elaborations on CS-IPM and IPM*

Broadly, CS-IPM is a process that can be realized by synchronizing knowledge about the biology of the targeted pests, the technology to curb these pests, and the pests' environment, while observing the minimum possible economic thresholds and damages to humans and the environment [4,5,10–17]. Egan et al. [4] more particularly define CS-IPM as an integrated approach that uses conservative and naturally eco-system compatible practices to minimize or stop pest insects, pathogens, and weeds to subsequently minimize farmers' dependence on chemical pesticides and thus damage to human and environmental health. Thus, CS-IPM practices comprise bio-pesticides, biologically-based pest control methods, the use of crop varieties, and other related practices [4,10,13,17–19]. CS-IPM practices embody an alternative that can sustainably and optimally minimize the use of chemical pesticides and their related toxicity to the environment resulting from pesticide overuse [4,10,20–23]. The benefits of the CS-IPM approach are therefore multiple and include savings in on-farm production costs, limited human, and animal exposure to pesticides, wider availability of organic foods, and eventually improved human health and environmental value [4,10,15,17,19,21–23]. Embracing climate-smart agricultural practices, innovations, and technologies has the desired potential to bring about the needed agricultural productivity, and eventually sustainable food systems [4,22,23]. Moreover, Climate Smart Agricultural Technologies (CATs) are also viewed as an up-to-date approach that can transform and reinvent agricultural production to achieve global food insecurity and nutrition [5,6,11,19,22,23]. An effective CS-IPM approach predicts changes in pest pressures caused by climate change and deploys effective monitoring, forecasting, and management actions to ensure improved farm outputs for people and the environment [3,4,10,19,23]. However, CS-IPM adoption and implementation have been low globally leading to adverse climate change effects in certain regions for instance northern Ghana [9,11,12]. The role of institutions, farm households' socio-economic context, the location biophysical, and characteristics of the new technology, among others, are vital determinants in the implementation and adoption of CSATs, including CS-IPM [4,5,11,17–23]. On the other hand, IPM, in general, focuses on a synchronized usage of various strategies to achieve optimal control of any pests (vertebrates, pathogens, or insects) while maximizing economic benefits and the wellness of the respective ecosystem [9,14,16,21,23]. Under IPM, various pests are targeted simultaneously and monitored consistently [4,14,16,21,23]. Under IPM, one must ensure that pesticides are used viably economically and that the strategies used to combat pests are well integrated either vertically (within certain pest types) or horizontally (across various pests) [4,9,14,16,21,23]. Hence, in a nutshell, IPM practices must generate economic benefits, minimize the usage of chemical pesticides, and preserve good human and environmental health. However, because such desired integration has been generally difficult to achieve across several farming systems, thus failing to maximize the desired and sustainable environmental and human health benefits, and the economic benefits, this has led to more targeted and efficient methods of pest management that maximize targeted environmental and economic benefits, for instance, CS-IPM [4,14,21,23]. We contribute to the literature by investigating important determinants for the deployment of CS-IPM practices that can enable communities to harness these integrated benefits.

The rest of the paper is organized as follows: next, we present the methods used to conduct this meta-analysis review, after which we present our results and discuss them. Finally, we conclude with guidance on feasible investment and policy alternatives.

## 2. Materials and Methods

### 2.1. Review Methodology

Our review adopted and followed the Preferred Reporting Items for Systematic Reviews and Meta-Analysis (PRISMA) methodology elaborated by Moher et al. [18] and Page et al. [19]. The PRISMA methodology of meta-analysis review follows a checklist of items that we considered throughout the review to improve the needed transparency while executing the review. Considered items comprise all features of a plausible manuscript

targeted for this review, for instance, the title of the manuscript, its abstract, its introduction, methods used in the manuscript, results reported, discussions of the reported results, and sometimes the funding sources. The objective of the PRISMA method of meta-analysis reviewing is to enable a thorough review of all possible original studies that empirically and soundly analyzed the subject matter, in this case, the CS-IPM socio-economic aspects globally. We show our simplified identification and selection criteria of considered studies in Figure 1.

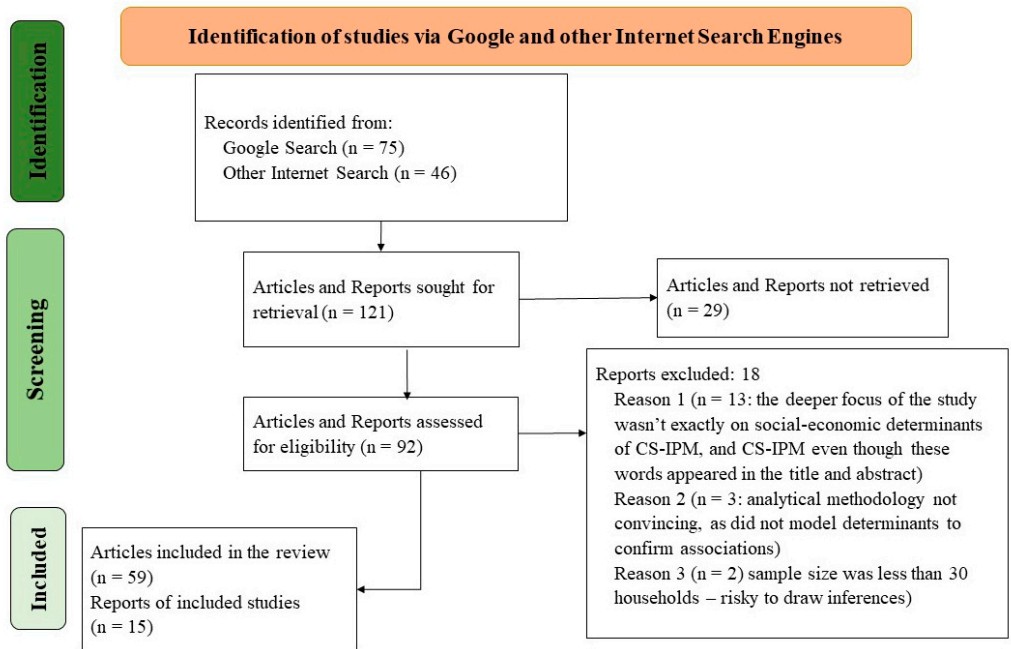

**Figure 1.** Identification and selection methodology of considered studies.

In selecting the papers considered for this review, we executed this rigorous, robust, and comprehensive meta-analysis literature review methodology, as has been recently applied by Gwara et al. [16] and Gwara et al. [17]. This methodology allows transparency and reproductivity. Specifically, the chosen PRISMA approach granted us the opportunity to execute a diverse and comprehensive review process, allowing us to smoothly synthesize research evidence in a reproductive manner [17].

Our focus was on farmers' populations, interactions of the population and CSATs, comparisons of CSATs and other conventional farming technologies, outcomes of adopted CSATs, including CS-IPM ones, and the study design being at least quantitative or qualitative. The intervention of focus was the "the determinants of implementation of CS-IPM in Ghana and Benin in comparison with other African countries and other cases in other parts of the world especially Asia". The outcome was the implementation of CSATs based on varied determinants with a bias in Ghana and Benin. Again, the designs of these studies were generally categorized as either qualitative or quantitative, while others employed a mixed set of methods.

## 2.2. Identification of Relevant Studies and Search Strategies

As a strategy, the study involved performing the primary search in the database of Google and other internet search engines using keywords such as "determinants of the implementation of CSATs specific to Ghana and Benin and then to the rest of Africa and Asia, climate-smart technologies, Integrated Pest Management technologies, and One health Innovations Ghana and Benin". This review was restricted to covering only peer-reviewed journal articles that were available and published in English but specifically addressed CSATs more so in the regions mentioned above. We also used the aspect of searching CSATs respective to integrated pest management (IPM) like "drought-resistant

seeds" to access more articles on respective practices. We more dominantly used a method of handpicking articles included in the electronic database of Google scholar about Climate Smart Agriculture to get more information on CSATs and related practices. Articles in Google Scholar are more complete empirically with proper abstracting, indexing, depth, and breadth of information [16,17]. We finally identified, through our Google and other internet search efforts, a total of 121 studies, but 29 were dropped, and we considered only 92 studies for this review (see Figure 1).

### 2.3. Screening Strategy and Employed Eligibility Criteria

The study sourced and screened 121 journal articles from the internet for their relevance to the objectives of this study, first based on the title. Those studies which satisfied the primary inclusion criteria were further finely vetted based on their abstracts. Subsequently, complete-text screening was only done for those studies that were relevant to the needed data. After this complete-text screening process, a total of 92 articles were identified as potential sources of information on CSATs as was needed by the review process, and finally, 74 were considered for this study (see Figure 1). More specifically, our review executed an approach that was typically descriptive analytically to source data from the selected articles that satisfied the inclusion conditions and standards. This was made more possible by following the framework elaborated and used by Gwara et al. [16,17]. Data extracted included names of the authors, year, location, the determinant of CSATs implementation studied, the study design used, study location, and the considered sample size. This study therefore gathered variables that give a qualitative assessment of the socio-economic determinants of CS-IPM technologies and practices, from the studies that were considered. Variables such as determinants of climate-smart agricultural technologies, CS-IPM, and Climate Smart Agriculture in general, were also considered.

### 3. Results and Discussions

In this section, we summarize key socio-economic factors that were dominantly reported in various studies to be influential in determining the successful deployment of CS-IPM practices. However, to make the understanding more simplified, we make an outline of these factors as sub-sections under Section 3, so that readers could easily distinguish them from each other, unlike if they were all documented in plain paragraphs.

### 3.1. Poor Access to Suitable Farming Equipment

The poor access to CS-IPM suitable farm tools is a substantial obstruction to deploying CS-IPM innovations in Africa [20,21]. CS-IPM innovations may not inevitably necessitate those farmers to acquire more farming equipment than that used in conventional agriculture, however, some needed equipment for CS-IPM may be particular but not readily available. For example, with respect to CSA where CS-IPM is a subdivision, certain conventional agriculture equipment can be used while others would need adjustment. For instance, hand hoes may need reductions in size to ably dig CSA compatible planting formations. On the other hand, insufficient inputs and farming materials, restricted access, and incapacity to afford better-quality seeds for crops and cover crops, inorganic fertilizers, and suitable pest management tools and products, do also hamper the deployment of CS-IPM innovations [20]. Moreover, proper CS-IPM improves yields by enabling biological procedures and farming practices that increase soil fertility, suppress pests, and improve weed control [14,22].

### 3.2. Access to Credit and Finance

Smallholder farmers targeting to adopt CS-IPM innovations and CSA practices, in general, are usually restricted by insufficient financial power that is needed to invest in needed capital for instance land, farming tools, improved seeds or breeds, labor, and other farm inputs. As noted by Milder et al. [20], CSA is therefore generally more gainful in the long-term in comparison to conventional farming. However, realizing these long-term

gains necessitates heavy preliminary investments that are usually risky, hugely expensive, and unaffordable to smallholder farmers. These smallholder farmers do particularly fear to take on such risks because it threatens their household food security. Hence, at the times when farmers are threatened with food security is threatened, they do usually give up on CS-IPM practices. Thus, long-term adoption of CS-IPM practices is more probable when CS-IPM delivers substantial paybacks in the initial one or two years of adoption [4,14,23]. However, these immediate gains are more probable when CS-IPM practices are employed alongside good agronomic practices and good-quality seeds. Most often, technology choices are restricted by a lack of financial resources and or awareness about how these resources could be accessed, if they were available. CS-IPM practices are therefore costly, and farmers must have seed money to use these practices or get subsidies to facilitate CS-IPM adoption [23]. The lack of sufficient financial support to obtain needed farm materials and inputs is therefore a significant impediment to SC-IPM adoption by smallholder farmers. For instance, usually more labor would be needed than what is possibly available, and the situation has been worsened by rural–urban migration of youths, the prevalence of HIV, other diseases, and malnutrition [24]. Therefore, regional financing for CS-IPM research, capacity building, and economic incentives that stimulate CS-IPM development and acceptance should be considered for the successful deployment of CSA in general [23].

### 3.3. Physical and Social Infrastructures

An enabling physical infrastructure that includes water supply and management facilities that can enable irrigation, transport and market facilities, communication, processing, and storage infrastructure is essential for smallholder farmers to be able to implement CS-IPM. Social infrastructure is also essential, for instance, for farmers' organizations. Poor and insufficient infrastructure limits the adoption of CS-IPM innovations, particularly for subsistence farmers, whose investment decisions rely on revenues generated from sales of their produce [25,26]. Moreover, the ecosystem of certain locations which comprise the location's people and the various activities they engage in, the location's topography, weather conditions, and general climate, and the location's political, economic, social, and cultural structure can effectively encourage or discourage adoption of given CS-IPM practices [27]. In addition, strategic discrepancies in market alignments (i.e., the blend of subsistence and commercial farming practices versus virtuously subsistence farming practices), and probable mitigation alternatives, for instance, crop intensification versus livestock farming, do all influence one's decision to apply CS-IPM practices [25].

On the other hand, non-physical barriers comprise those factors that affect the environment desirable to practice CS-IPM. These are usually institutional elements that comprise a steady macroeconomic environment, functional institutions and policies, provision of incentives through markets, and leadership structures that facilitate the generation, dissemination, and usage of CS-IPM relevant technologies, practices, innovations, approaches, information, knowledge and skills, and public and private investments in human, social, natural, and capital [28]. In a community where institutions, policies, and markets are not effective, the generation and dissemination of information, technologies, and innovations become crippled, hence leading to farmers' deficient awareness, knowledge, and skills to adopt pro-CS-IPM practices [25]. Additionally, obstructive land tenure systems, a lack of lucrative markets and safe post-harvest storage, meager financial services, frail farmer organizations, and minimal household incomes render it risky and unbearable for farmers to change from any of the conventional practices to CS-IPM ones [29].

### 3.4. Access to Appropriate Technologies and Technological Diffusion

Adaptive strategies that are essential in managing climate change effectively require appropriate practices, technologies, and innovations. For instance, the IPCC [30] denotes that the absence of suitable technologies can seriously limit a community's capability to implement CS-IPM innovations by reducing alternative choices of probable interventions.

Adaptive strategies and capacities are likely to differ, based on access to a given technology at different levels of the target sectors. Different levels and stages of the technology's adoption in communities and the communities' ability to adapt this technology to the local context are key determining factors of the community's capacity to change and accept this technology. Hence, community awareness of the new technologies and the utilization of these technologies are important in cementing the adaptive capacity of these communities to CS-IPM innovations. The technology's expansion and diffusion in communities are key worries associated with low CS-IPM adoption. Moreover, the sluggish distribution rate and adaptation of CS-IPM in Africa are partly attributable to low technology adoption [31]. In fact, the role played by extension services in diffusing technologies to farmers is hindered by farmers' poor financial capabilities, low technical skills, and threats of the technology that are interruptive to the prevailing household livelihood systems [32]. For example, to avert disease transmission, farmers put tree-crop defense barricades around cocoa farms. These disease barricades effectively minimize Cocoa swollen shoot virus disease infection rates [33]. Even though these barricade plantings take up land area, they are also a source of income working as an agroforestry extension where farmers grow crops like rubber, citrus, palms, etc. which all enhance effective CS-IPM practices. Therefore, monitoring tree health can effectively help farmers avoid major pest and disease occurrences [34]. Hence, appropriate capacity building of farmers through demonstration farms, appropriate extension services, and consideration of indigenous knowledge and feasible cultural practices are essential for the meaningful technological transfer, which extends to CS-IPM innovations.

### 3.5. Benefits from the Technology

The technology must be widely beneficial to households to foster its uptake. However, all components of benefits are hard to embody in each technology. For instance, appropriate technologies must be catering to ecological, economic, and social benefits to be widely adopted and accepted [7,35–37]. For instance, farmers in SSA usually prefer technologies that fetch immediate and significant benefits especially economically to those whose benefits are achieved in the future because subsistence farmers face immediate food security risks and high costs of production [38]. However, a lumpsum of the benefits of CSA, including CS-IPM practices, can mostly be attained in the longer term [11,39]. Moreover, no analyses of costs and benefits for CS-IPM has been done yet but given that it is rated more complex than IPM in some instances, related CS-IPM cost, especially in the starting phases, may be high, hence threatening the immediate economic benefits targeted by most farmers [40,41]. For instance, evidence shows that IPM practices improve crop yields to about 41% and reduce pesticide use by about 31%, which all increases the net income received by farmers [42,43]. Therefore, to enhance their uptake CS-IPM innovations must be beneficial to adopters especially regarding economic returns and food security sustainability. Nevertheless, the projected benefits of CS-IPM innovations would generally be high due to reductions in external negatives [40,41]. Negative external costs can arise from contamination of groundwater caused by heavy pesticide use [44] or severe pesticide poisoning that affects 1–3% of farmworkers globally, leading to numerous deaths. Moreover, even though farm households in low developed countries use about only 25% of global pesticides, these suffer about 99% of deaths from pesticide poisoning [45]. Other costly negative externalities include the unintentional killing of soil and water organisms that predate pests, soil and water pollution, and the death of several flora and fauna. Therefore, such external negative costs must be avoided for any CS-IPM innovation to be successful [11,46].

### 3.6. Poor Leadership, Management, and Governance Structures

Leadership, governance, and management infrastructure reflects, influences, and guides the legal framework that shapes the socio-economic landscape of a society [32,47]. Bad governance is a key inhibitor of the desired socio-economic developments that must foster the society's adaptation and adoption of CS-IPM. Incompetent societal management

usually focuses on instant gains, thus those innovations needing heavy investments with later gains, for instance, CS-IPM, may not be prioritized [32]. Incompetent leadership may be characterized by corruption and failed systems and usually fails to guarantee institutional development and effectiveness which limits access to resources needed by communities to realize CS-IPM innovations [47,48]. Therefore, poorly managed institutions fail to mitigate climate risks, and thus deployment of CS-IPM innovations and institutional deficiencies lower communities' abilities to combat vulnerability to climate change [49]. Therefore, necessary improvements in policies and societal management are needed to make agricultural systems more compatible with CS-IPM innovations and thus resilient to changes in climate [50]. However, building suitable institutional adaptive capacity necessitates good coordination, empirical understanding of societal problems, candidness in addressing societal challenges, candidness in exploiting opportunities, aggressive community participation, and solid political will [51,52]. Therefore, for successful deployment of CS-IPM, critical elements like setting up a decent agenda for collaboration, capacity building, community involvement, raising awareness, rational investments, and suitable technological and governance structures must be available [32]. Unfortunately, Africa, where a large proportion of vulnerable smallholder farmers are, has generally inadequate supportive governance structures. Had they been adequate, they would have been appropriate pathways for providing suitable extension facilities, credit services, fairness in the allocation of resources, effective information diffusion, a decent policy environment, and functional social institutions that all foster deployment of CS-IPM innovations [53].

### 3.7. Policy Issues

In West Africa, there is yet to be a comprehensive policy and a robust institutional framework on adaptation to climate change and CS-IPM innovations, especially by smallholder farmers. Improving policies and regulations as is projected, for instance, in Ghana, enables the production and certification of biopesticides and other biologically-based pest control products that are important nature-dependent solutions within the general CS-IPM innovations framework [47]. However, there are policies on various domains such as land and water resources management which could somehow enable the deployment of CS-IPM innovations [32]. Thus, unless the current gaps in concerned government policies are addressed, it will be tough to successfully deploy CS-IPM innovations, and CSA in general [32,47]. Moreover, different sector-based policies must be coordinated and integrated to echo how the inter-linked sectors would address climate change, and thus deployment of CS-IPM innovations. Nevertheless, there has been some good steps to address these inconsistencies. For instance, many governments in Africa have agricultural plans that integrate and prioritize a climate change focus, for example the National Agriculture Investment Plans. However, these agricultural plans are hardly scientifically evaluated to inform their implementation, leading to a poor understanding of the bio-physical, organizational, economical, and ecological barriers to the deployment of CS-IPM innovations, thus crippling the success of CS-IPM innovations in both farm and food system [25]. Therefore, policies must be coordinated across sectors (within the private and public arenas) and cover a wide spectrum of key aspects surrounding the deployment of CS-IPM practices to achieve their adoption at scale. However, good funding and suitable expertise are essential to frame suitable CS-IPM plans and policies [45]. On the other hand, formidable community support, awareness-raising, and knowledge transfer are essential for the realization of appropriate up-to-date and harmonized policies that enable CS-IPM deployment [25,45]. For example, communities of cotton farming households in northern parts of Benin willingly pay annually a substantial amount per hectare for the diffusion of CSA extension services that include CS-IPM innovations [54]. Moreover, most cotton farming households in Benin, even though they are largely illiterate and vastly poor, were willing to pay for training and regular advisory extension services on CS-IPM innovations due to these services' significant net financial and environmental advantages.

### 3.8. Hostile Land Tenure Systems

Land tenure refers to a system of land use rights and legally acceptable institutions that regulate access to land and land use [55]. Land tenure systems that are vague, casual, or overlapping do constrain individuals' investment interests in sustainable water and soil management (which includes CS-IPM innovations). For instance, customary land tenure systems usually present barriers to the implementation of CS-IPM innovations [20]. These barriers may include one's shorter duration of access to land, yet CSA in general, including CS-IPM practices, usually requires long-term investments in soil fertility and land resource sustainability in general. Therefore, clear, and long-term individual farm households' access and proper user land rights would foster continuous and long-term investments, thus effectively enabling adoption and deployment of CS-IPM innovations [56,57]. Investments in improved CS-IPM necessitates protected land tenure systems as a precondition to invest in climate change adaptation approaches like CS-IPM [56]. For example, in Uganda's Mubuku irrigation scheme, Ngigi [32] found that there was a rampant withdrawal of development partners from the support of the scheme, because of land tenure systems that were insecure. Insecure land tenure would condemn renting smallholder farmers to restricted land use rights, thus having minimal authority on their favorable investment decisions on the land.

### 3.9. Cultural Limitations

Cultural values avail both opportunities as well as barriers to communities' acceptance, adoption, and diffusion of innovations, including CS-IPM ones. For instance, Nielsen and Reenberg [58], in their study of two comparable ethnic groups in Burkina Faso, where Fulbe and Rimaiibe discovered that the Rimaiibe easily accepted diversifications in their livelihood approaches to adjust to climate change, however, the Fulbe instead found the same approaches unacceptable. Such insights throw light on how societal culture may render certain adaptive approaches to climate change impractical even though these have been acceptable in other cultures. Therefore, understanding well the context of the local culture is valuable for the successful deployment of CS-IPM innovations. Discussions about cultural practices and beliefs are also thus essential [58]. This is because there is a necessity to study the behavioral, institutional, and cultural barriers to consider their complementarity with the proposed innovations [6,15]. Smallholder agriculture in SSA is a "way of life" within which several cultural practices, norms, and beliefs are an important life aspect [58]. Culture, beliefs, and behavior govern individuals' actions, acceptable practices, privileges, and rights in food production, distribution, and consumption activities, thus must be considered in the deployment of CS-IPM innovations. However, certain cultural activities like belonging to social groups instead boost knowledge acquisition and sharing, thus fostering the adoption and deployment of CS-IPM [6,15]. Nevertheless, there are many culturally motivated factors, including inadequate access to financial support, especially for women, yet costs are incurred in the deployment of CS-IPM which hinders CS-IPM deployment [11].

### 3.10. Gender Inequalities

In terms of gender dimensions, women's role in agriculture, for instance in Benin, Ghana, and the rest of SSA, is substantial. In Benin, women make up to 35 percent of employment in agriculture, and 14 percent of agricultural households are led by women [54]. However, despite their contributions to the sector, women experience several challenges in water and land use including restricted land access and user rights, again due to cultural customs [59]. Moreover, women also face more obstacles than men in accessing improved inputs, agricultural production technology, land, education and extension services, and finance, which are all essential in the deployment of CS-IPM innovations [12,59–62]. The shortage of productive capital to women lessens this proportion of farmers' capacity to deal with adverse climate impacts, and thus deployment of CS-IPM innovations [63]. Hence, gender issues can form the required background for prioritization and adoption,

and deployment of CS-IPM technologies, and practices. For instance, in Lesotho, females who modify farmlands that they are not culturally permitted to possess may lose these farms so that farms are redistributed to their masculine relatives [62]. Furthermore, in some parts of Ghana, women can only access lesser land for agriculture, which renders trying new innovations like CS-IPM riskier, more so that CS-IPM outcomes may not yet be confirmed [56]. Therefore, gender-responsive planning is critical for the success of innovations, including CS-IPM [64].

*3.11. Knowledge on CS-IPM Practices*

An integrated and effective approach to controlling pests requires good technical awareness and knowledge which necessitates continuous monitoring, thus rendering the approach more expensive with multiple indirect costs, unlike the plain usage of pesticides. Therefore, technical awareness and advocacy aimed at promoting CS-IPM innovations and the concept at large is indispensable and can only be achieved through an effective knowledge transfer process [32,65]. For example, deficiencies in knowledge have been found to be key impediments to innovation uptake among Ghana's farming households [29]. Unfortunately, most farm households in Africa either lack proper access to knowledge or sometimes are unable to utilize effectively existing knowledge [65]. Moreover, successful and useful adaptation to innovations necessitates the need to have sufficient knowledge about existing alternatives of the innovations to be adapted, the technical ability to evaluate these alternatives, and subsequently the technical capability to select contextually suitable alternatives for implementation [66,67]. With regards to climate change, the need for knowledge can be revealed through the acquisition and diffusion of appropriate information, for instance on weather hazards, diseases, and pests. Lack of knowledge on CS-IPM due to low technical and logistical capacity of rural extension agents impedes creation of awareness on CS-IPM innovations and removes any incentives to deserving farmers to adopt CS-IPM [14,67]. Lack of CS-IPM knowledge has instead increased pesticide application as the principal strategy against climate-induced pests [20,68,69]. Moreover, proper knowledge improves the identification of new and invasive pest species and allows the good predictive ability for appropriate CS-IPM innovations against emerging pests species. Proper knowledge has, for instance, allowed effective control of the invasive fall armyworm [70,71].

Therefore, overall knowledge processes for a successful operationalization of climate smart IPM include but are not limited to the following elements:

- Climate-informed models of pest risks and candidate natural enemies, forecasting and monitoring, technical advisory extension services and protocols, and priority resetting of management alternatives [72–74].
- Timely detection of invasive species and preventive action against future climate-risk pests.

This implies that there is a need for an efficient implementation support strategy and partnerships to enable:

- Enhancing governmental pest management front agents, farmers, and other next- and end-users' capabilities in reporting, anticipation, proactiveness, and response.
- Finetuning pilot evidence-based innovations and fostering the use of digital tools including Apps-led pest scouting and warning devices.
- Co-creating business models for pest management services and engaging private sector for sustainable deployment of impactful products and tools by empowering champion youth and women farmers.
- Policy makers' engagement to trigger a legislative enabling environment and coercive measures for example against abusive/misuse of chemicals and particularly those that are prohibited and are of high toxicity to nature and having significant non-target negative effects.
- Accelerate the co-development and coordination of functional local and regional early warning and rapid response systems against pests' invasions.

However, these knowledge processes and components must be hinged on an efficient regulatory system that phases out, in a timely manner, highly hazardous pesticides (HHPs) and active ingredients that pests have developed resistance to [72–74].

### 3.12. Extension Services Systems' Infrastructure

Increasing demand and limited resources have put an enormous strain on much of SSA's extension services infrastructure, leading to a degradation of their services. However, an appropriate, effective, responsive, accessible, and well-informed extension system is indispensable for supporting farmers to a successful adoption and implementation of CS-IPM. Conversely, extension services systems in many developing countries are deficient of these features, hence limiting farmer's adoption of CS-IPM [14]. These deficiencies are due to illiteracy of extension agents about climate change, limited funding, lack of sufficient qualified personnel to deliver the extension services, poor coordination among key stakeholders for a given innovation for instance farmers, researchers, policy makers, extension agents, and remote localities of farm households [14,69,75–77]. Remarkably, weather centered advisory extension services are already occurring for various pests [74,78], although their employment in developing countries remains challenging. Hence, improved communication between countries and regions is desired, to implement applicable CS-IPM practices at regional level to effectively contain possible pests' outbreaks [45,79]. Moreover, about 70% of the predicted climate possibilities for the future are already occurring, thus adjusting to a 'climate analogues' (a web application that shows two or three or more places and their respective climates, that individually have analogous climates for a given day as that of a place specified by the user in the future) method spearheaded by CGIAR is contextually valuable [80,81]. Therefore, escalating this method or approach to include pests would be pivotal to understanding what species of pests could be challenging for crops, localities, and cropping patterns, henceforth informing appropriate changes in extension services systems [82]. However, extension services would require robust connections to reliable diagnostic services to guarantee that pests diagnoses done in the field are accurate for appropriate recommendations [14,74,83,84], and that CSATs, including CS-IPM practices, are optimally applicable [85–94].

In a nutshell, we clearly identify important social-economic determinants for the deployment of CS-IPM practices as elaborated above, and we highlight some of these factors in Table 1. Consideration of these factors in any CS-IPM initiative has been widely found in the literature reviewed above, to improve the adoption and deployment of CS-IPM innovations and practices.

**Table 1.** Summary of some of the key studies reviewed.

| Reference | Country | Sample Size | Data Type | KEY CSA—IPM Studied | Summary of Key Findings | Analytical Method Used |
|---|---|---|---|---|---|---|
| Sain et al. [11] | Guatemala | 42 CSATs stakeholders and 200 farms. | Survey Data | Pest and disease tolerant variety. | • The CSATs analyzed are profitable except stone barriers.<br>• Cost and benefit analysis of CSATs mainly concerned the implementation of selected CSATs.<br>• The practice largely requires 4 to 9 years to generate economic gains. | Mixed |

**Table 1.** *Cont.*

| Reference | Country | Sample Size | Data Type | KEY CSA—IPM Studied | Summary of Key Findings | Analytical Method Used |
|---|---|---|---|---|---|---|
| Chhetri et al. [3] | India | 16 villages in 4 diverse rainfall zones | Survey data. | Rainwater harvesting and storage. | • The farmers' age significantly affects the grading of CSATs. <br>• Feale farmers prefer integrated pest management, weather-centered crop advisories services, and contingent crop management compared to males. <br>• The results showed that farmers' preference for CSATs significantly differs according to rainfall zones. | Mixed. |
| Shirsath et al. [88] | Bihar-North-Eastern India. | 194 land units from 34 zones. | Survey data. | Seed replacement. | • Input intensification as well as CSATs in current and future climate. <br>• Present technologies will not be climate-smart for the coming years. <br>• Even though present land use practices will not be climate-smart for the future, novel technological intercessions will aid abate their negative impact. | Spread sheet-based methodology. |
| Atitianti et al. [1] | Ghana. | 80 Households. | Survey data | Using uprooted weeds and other biomass for mulching cleared lands. | • Higher financial capital for CSATs farmers in their average incomes as compared to conventional farmers. <br>• Farm tenure determines CSATs practices. <br>• Increasing farmers' age (to a given threshold), was associated with increased adoption of CSATs practices. | Mixed method. |
| Anuga et al. [9] | Ghana | 320 smallholder farmers. | Survey data. | Mulching to avoid excessive use of water. | • Social and cultural factors strongly influence CSATs' adoption by farmers. <br>• Institutional factors positively predict CSATs adoption. <br>• Personal aspects do not influence adoption of CSATs. | Mixed method. |
| Zakaria et al. [6] | Ghana | 543 Rice farmers. | Quantitative. | Improved seed use. (Drought-tolerant rice) | • Farmers that have off-farm opportunities are more likely to accept CSATs. <br>• Farmers that have small farm sizes would accept CSATs if costs are minimal. <br>• Access to in-put credit besides engagement in contract farming and production credit influenced the adoption of CSATs. | Mixed using Quantitative Multivariate Probit Model. |
| Heeb et al. [14] | No specific country. | | Descriptive. | Pest populations could be prevented by applying multiple pest preventing methods. | • CSA-IPM must broadly be addressed at all levels other than doing so only at farm level but in coordination with evidence-based policies. <br>• There should be assessment, selection, and evaluation of identified CSA-IPM options at all levels. | Review. |

**Table 1.** *Cont.*

| Reference | Country | Sample Size | Data Type | KEY CSA—IPM Studied | Summary of Key Findings | Analytical Method Used |
|---|---|---|---|---|---|---|
| Issahaku et al. [40] | Ghana | 476 households. | Survey | Drought tolerant and early maturing varieties. | • CSATs adoption significantly improves nutrition and food security.<br>• The impact of adoption differs across quartiles of nutrition and food security outcomes, and agro-ecological zones.<br>• Proper understanding of the drivers of CSATs' adoption improves the designing and diffusion strategies of CSATs thus increasing CSAT's adoption. | Mixed and also using Endogenous switching regression. |
| Issahaku et al. [41] | Ghana. | 476 households. | Survey | Crop choice. | • Household characteristics and endowments influence household decisions to adopt CSATs.<br>• Access to inputs, herbicides and fertilizers positively influences households' execution CSATs.<br>• Herbicide's usage is gaining in importance for Ghanaian smallholder farm households. | Mixed |
| Krishna et al. [85] | Ethiopia. | 380 households. | Survey data | Change of crop variety. | • Smallholder farmers' access to affordable credit services is key for CSAT's adoptability.<br>• Protected land use rights are significantly motivated investments in viable CSATs adoption.<br>• The off-farm income positively influences the decision of smallholder households to adopt CSATs. | Mixed. Used primary and secondary sources of data. |
| Kangogo et al. [21] | Kenya. | 792 potato farmers. | Survey data. | Irrigation. | • Adoption of all CSATs remains low and stands at only 45% in developing countries.<br>• There is also interdependence between practices, for example, certified seed and irrigation extra.<br>• CSATs that require intensive skills and financial investments are associated with risk-taking among farmers. | Mixed. |
| Zakaria et al. [15] | Ghana. | 300 farmers | Survey data | Changing planting periods. | • Access to credit positively influences SCATs adoption, but only 15% of smallholder farmers do have access to agricultural credit.<br>• Only 11% of farmers do insure their crops against uncertainties despite farm insurance being introduced.<br>• Averagely, smallholder farmers accept up to around five CSATs practices to improve agricultural production and productivity. | Mixed |

**Table 1.** *Cont.*

| Reference | Country | Sample Size | Data Type | KEY CSA—IPM Studied | Summary of Key Findings | Analytical Method Used |
|---|---|---|---|---|---|---|
| Chanana et al. [12] | India | 375 households. | Survey data | Change sowing dates. | • Gender disaggregated data is so important in prioritizing and deploying CSA innovations, technologies, and practices. <br> • Supportive institutions are central in realization of gender-sensitive implementation of CSATs. | Mixed |
| Christie et al. [10] | Ghana and Mali | 293 respondents survey | Survey data | Pesticides use. | • Gender-inclusive designs are strongly associated with good in Ghana and Mali <br> • Men are usually in charge of applications of pesticides on farms. | Mixed method. |
| Houngbo et al. [67] | Benin. | 1237 maize farmers. | National survey data. | Single cropping in rotation with other crops. | • Use of synthetic pesticides was the most single control practice used by farmers. <br> • Neem leaves or seeds are used to locally produce botanical pesticides by smallholder farmers. <br> • Management practices exhibited by farmers are importantly significantly related to farmers' knowledge of managing pests. | National survey on maize farmers. |
| Loko et al. [65]. | Benin | 83 producers of Kersting's groundnuts. | Survey data. | Use of chemicals. | • The attack of insects was the most devastating of the four storage constraints identified. <br> • Promotion of Kersting's ground nuts varieties can help minimize storage constraints. <br> • The palatability of stored Kersting's groundnut seeds attract attacks from insect pests. | Mixed |
| Ratnadass and Deguine [7]. | No specific country | Not mentioned. | Review data | Use of synthetic insecticides and rodenticides. | • Crop protection practices in the paper were applied mainly to pests. <br> • Agroecosystem redesign is the most appropriate that is likely to foster enhancement of the One Health Notion. <br> • The Agroecological Crop Protection framework addresses major global challenges like climate resilience, among others. | Review methodology. |
| Makate [47] | Sub-Saharan Africa | Not indicated | Review data | Use of small-scale irrigation. | • The approaches or strategies identified must boost innovative technologies in agriculture include the climate-smart village approach extra. <br> • Better markets and market information access positively drive demand upwards for CSATs like CS-IPM thus promoting adoption of CSATs. <br> • Functional institutions and good policies are central to successful scaling of CSATs. | Institutional analysis and development (IAD) Framework method. |

**Table 1.** *Cont.*

| Reference | Country | Sample Size | Data Type | KEY CSA—IPM Studied | Summary of Key Findings | Analytical Method Used |
|---|---|---|---|---|---|---|
| Egan et al. [4] | Sub-Saharan Africa | Not mentioned | Review data | Host plant resistance and tolerance. | • Practical implementation of Climate Smart IPM requires; climate-informed advisory services,<br>• Participatory research for CS-IPM,<br>• and policy development. | Technical Reporting. |

## 4. Conclusions

In this review, we try to understand socio-economic factors that determine the adoption, implementation, and successful deployment of CS-IPM technologies and practices. Generally, adoption of climate-smart innovations, practices, and innovations that enable an ecological and sustainable management of pests is low, with most farmers relying on more and more use of pesticides as the major option against pests' populations that are climate-induced or otherwise. The more frequently stressed factors that deter immediate and extensive adoption of sustainable CS-IPM innovations are largely farm-level issues that include the gender, education level, and age of household heads, household farm size, income, access to markets and finance, and location. However, institutional barriers like lack of appropriate extension services, poor research infrastructure, and antagonizing policy frameworks do also persistently frustrate and fail adoption of CS-IPM innovations and CSA in general. Addressing these factors concurrently, therefore, would significantly pave a clear way for willing farm households to adopt CS-IPM innovations, thus fostering a more sustainable and extensive scaling of multiple CS-IPM practices. A clear approach that harmonizes and conveys together the recognized CS-IPM practices and tools with a robust emphasis on adjusting these tools and practices to all changes that are climate-induced while prioritizing pests, is therefore, very crucial for the successful and scaled adoption and implementation of CSATs and CS-IPM. This approach will require a multi-stakeholder platform strategy that involves public support, and political leadership that is embedded within a favorable enabling CS-IPM implementing environment. Without proper coordination of the several stakeholders needed for effective adoption of innovations, the large-scale uptake of novel and appropriate pest management innovations, practices, methods, approaches, and technologies will only yield inefficient and vastly unsuccessful efforts of CS-IPM deployment. Moreover, CS-IPM is a dynamic and an ever-evolving approach responding to changes in the climate. Therefore, continuous monitoring and evaluation of activities and interactions of all stakeholders are required to assess and appropriately guide the execution of CS-IPM interventions, and their impacts. Such assessment enables continuous re-evaluation of the tools, practices, and approaches for the selection of those that suit the given context best. Similarly, effective communication on pest pressures through data sharing between national and international institutions is pivotal not only in enhancing knowledge on the implementation of appropriate CS-IPM practices, but also to foster connection across nature-based solutions and CS-IPM deployment. However, the CS-IPM implementation process should be bottom-up, which employs participatory prioritization based on farmers' indicators, but is also appropriate to the bio-physical and socio-economic context.

**Author Contributions:** Conceptualization, H.S. and G.T.T.-Y.; methodology, H.S., Y.K. and S.P.D.; software, H.S., Y.K. and S.P.D.; validation, H.S., G.T.T.-Y., R.D., V.C., C.G. and M.T.; formal analysis, H.S., Y.K. and S.P.D.; investigation, H.S., Y.K. and S.P.D.; resources, G.T.T.-Y., R.D., V.C., C.G. and M.T.; data curation, H.S., Y.K. and S.P.D.; writing—original draft preparation, H.S., Y.K. and S.P.D.; writing—review and editing, H.S., G.T.T.-Y., R.D., V.C., C.G., Y.K., S.P.D. and M.T.; visualization, H.S., G.T.T.-Y., R.D., V.C., C.G. and M.T.; supervision, H.S., G.T.T.-Y., R.D., V.C., C.G. and M.T.; project administration, H.S., G.T.T.-Y., R.D., V.C., C.G. and M.T.; funding acquisition, G.T.T.-Y., R.D., V.C., C.G. and M.T. All authors have read and agreed to the published version of the manuscript.

**Funding:** This research was funded by the World Bank via the Accelerating Impacts of CGIAR Climate Research for Africa (AICCRA) project, grant number AICCRA, P173398 and The APC was funded by World Bank.

**Institutional Review Board Statement:** Not applicable.

**Informed Consent Statement:** Not applicable.

**Data Availability Statement:** This is a meta-analysis from already published works that are effectively referenced and available on the internet.

**Acknowledgments:** We acknowledge the funding received from the World Bank to the Accelerating Impacts of CGIAR Climate Research for Africa (AICCRA, P173398) project. The AICCRA project is supported by a grant from the International Development Association (IDA) of the World Bank. IDA helps the world's poorest countries by providing grants and low to zero-interest loans for projects and programs that boost economic growth, reduce poverty, and improve poor people's lives. IDA is one of the largest sources of assistance for the world's 76 poorest countries, 39 of which are in Africa. Annual IDA commitments have averaged about $21 billion over circa 2017–2020, with approximately 61 percent going to Africa. We also acknowledge the tremendous review efforts from the three anonymous reviewers.

**Conflicts of Interest:** The authors declare no conflict of interest.

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
