# Peer review of "Determinants for Deployment of Climate-Smart Integrated Pest Management Practices: A Meta-Analysis Approach"

_agriculture, doi:10.3390/agriculture12071052_

Round 1
Reviewer 1 Report
The narrative is better now, but still, there are portions that are confusing, making it difficult to follow the idea or concept being developed. For example, the introduction is verbose and indirect. I edited the beginning of the introduction, from line 45 to line 67:
"Many vulnerable populations around the globe depend on agriculture for subsistence [1]. Although improvements in agricultural productivity in recent decades have enabled considerable reductions in global hunger [2], the agricultural sector still faces many challenges including trade barriers, scarcity of resources, breakdown of market systems, unstable and ineffective socio-economic policies, insecurity, increasing population pressure, unsustainable and soil degrading agronomic practices, the dilapidation of the environment, weather unpredictability and climate change [3]. These barriers to agricultural production risk food security and safety of rural populations, especially of the resource-poor smallholder farmers [4]. The adoption of climate-smart practices might help mitigate these risks [3, 5]. Climate-smart practices are often combined with Integrated Pest Management (IPM) practices that foster the rational and parsimonious use of pesticides, thus reducing dependence on conventional chemical pesticides[4]. Climate Smart Agriculture (CSA) is addressing these challenges by helping the agricultural sector to adapt to climate change and reduce emissions, concepts that are receiving wider global endorsement and are progressively gaining in importance, especially in strategic farming systems such as the cocoa production system in Ghana [1, 3, 6]. CSA aims to optimize social, environmental, and economic benefits while sustainably bringing development by anchoring on three key strategic pillars: (1) sustainably improve agricultural productivity and household income; (2) adjust to and build formidable household resilience to climate change, and (3) minimize and/or eliminate greenhouse gases emissions if compared to conventional agricultural practices."
The manuscript should be thoroughly edited to make the text more direct and concise, so that individual ideas, concepts, and conclusions are easier for the reader to follow.
For example, here are some exerpts and my comments:
398 Severally, however, these agricultural plans are hardly scientif-
399 ically evaluated as business cases.
There’s no need for the word “Severally”
What does this actually mean? “Scientifically evaluated as business case”
424 There-
425 fore, clear, and long-standing individual farm households’ access and proper user land
426 rights would foster such continuous and long-standing investments, thus effectively ena-
427 bling adoption and deployment of CS-IPM innovations [56, 57]. Therefore, investment in
428 improved CS-IPM necessitates protected land tenure systems as a precondition to contin-
429 uous and significant investments in climate change feasible adaptations and thus CS-IPM
430 innovations related to water and land management [56].
There’s no need for the word “Therefore” in 424/425 and 427. “Therefore” implies a conclusion based on the previous statement, and what we have here are two statements (424/425 and 427).
455 Nevertheless, many including inadequate access to monies especially for
456 women, yet costs are incurred in the deployment of CS-IPM hinders CS-IPM deployment 457 [11].
I don’t understand the logic of this statement (455-457)..
535 Remarkably,
535 weather centered advisory extension services already occurring for various pests [78, 79],
536 although their employment in developing countries remains challenging. Therefore, im-
537 proved communication between countries and regions is desired, to implement applicable
538 CS-IPM practices at regional level to effectively contain possible pests’ outbreaks [45, 80].
539 Moreover, about 70% of the predicted climate possibilities for the future are already al-
540 leged to be occurring on earth in some places, thus adjusting a ‘climate analogues’ method
541 spearheaded by CGIAR is contextually valuable [81, 82]. Therefore, escalating this method
542 or approach to include pests would be pivotal to understanding what species of pests
543 could be challenging for crops, and localities, cropping patterns, and henceforth inform-
544 ing appropriate changes in extension services systems [83].
I find the above difficult to understand, and here are my comments:
“Therefore” implies a conclusion based on the previous statement, a what we have here is a non-sequitor. (538)
What does this actually mean? “About 70% of predicted climate possibilities” (539)
The language is too diffused and inprecise. What does this actually mean? “are alleged to be occurring on earth in some places” (540) [Alledged?] [on earth? if not on earth, where else?] [in some place?]
What is the ‘climate analogues’ method? “(540)
Is climate analogues a method or an approach? (541/542)
“Therefore” implies a conclusion based on the previous statement, a what we have here is, again, a non-sequitor. (541)
563 what does "sustainable management of pests in low, " means?
The revised text is much better but certain sections are still hard to follow. I suggest that the entire document be edited by a native English language technical writer, to make the text more direct and concise, so that ideas, concepts, and conclusions are easier for the reader to follow. Also to correct misspellings, grammar and style.
Author Response
REVIEWER ONE
The narrative is better now, but still, there are portions that are confusing, making it difficult to follow the idea or concept being developed. For example, the introduction is verbose and indirect. I edited the beginning of the introduction, from line 45 to line 67:
"Many vulnerable populations around the globe depend on agriculture for subsistence [1]. Although improvements in agricultural productivity in recent decades have enabled considerable reductions in global hunger [2], the agricultural sector still faces many challenges including trade barriers, scarcity of resources, breakdown of market systems, unstable and ineffective socio-economic policies, insecurity, increasing population pressure, unsustainable and soil degrading agronomic practices, the dilapidation of the environment, weather unpredictability and climate change [3]. These barriers to agricultural production risk food security and safety of rural populations, especially of the resource-poor smallholder farmers [4]. The adoption of climate-smart practices might help mitigate these risks [3, 5]. Climate-smart practices are often combined with Integrated Pest Management (IPM) practices that foster the rational and parsimonious use of pesticides, thus reducing dependence on conventional chemical pesticides [4]. Climate-Smart Agriculture (CSA) is addressing these challenges by helping the agricultural sector to adapt to climate change and reduce emissions, concepts that are receiving wider global endorsement and are progressively gaining in importance, especially in strategic farming systems such as the cocoa production system in Ghana [1, 3, 6]. CSA aims to optimize social, environmental, and economic benefits while sustainably bringing development by anchoring on three key strategic pillars: (1) sustainably improve agricultural productivity and household income; (2) adjust to and build formidable household resilience to climate change, and (3) minimize and/or eliminate greenhouse gases emissions if compared to conventional agricultural practices." We are grateful to this example of how best we can make the manuscript more concise. We adopt this approach and exercise it over all the manuscript – in addition, we adopt the changes fully as suggested by the reviewer from lines 45 – 67.
The manuscript should be thoroughly edited to make the text more direct and concise, so that individual ideas, concepts, and conclusions are easier for the reader to follow. We accept and execute this guidance all through the paper, making it less wordy, direct and concise.
For example, here are some exerpts and my comments:
398 Severally, however, these agricultural plans are hardly scientif-
399 ically evaluated as business cases.
There’s no need for the word “Severally”
What does this actually mean? “Scientifically evaluated as business case”
The above suggestions have been incorporated and redundant working removed from the paper, replaced with more direct and concise statements
424 There-
425 fore, clear, and long-standing individual farm households’ access and proper user land
426 rights would foster such continuous and long-standing investments, thus effectively ena-
427 bling adoption and deployment of CS-IPM innovations [56, 57]. Therefore, investment in
428 improved CS-IPM necessitates protected land tenure systems as a precondition to contin-
429 uous and significant investments in climate change feasible adaptations and thus CS-IPM
430 innovations related to water and land management [56].
There’s no need for the word “Therefore” in 424/425 and 427. “Therefore” implies a conclusion based on the previous statement, and what we have here are two statements (424/425 and 427).
Advice considered and redundant use of the term “therefore’ is eliminated in the revised version
455 Nevertheless, many including inadequate access to monies especially for
456 women, yet costs are incurred in the deployment of CS-IPM hinders CS-IPM deployment 457 [11].
I don’t understand the logic of this statement (455-457).. The statement is re-written to clear state that since women have restricted access to finances, usually this can restrict their adoption of CS-IPM practices.
535 Remarkably,
535 weather centered advisory extension services already occurring for various pests [78, 79],
536 although their employment in developing countries remains challenging. Therefore, im-
537 proved communication between countries and regions is desired, to implement applicable
538 CS-IPM practices at regional level to effectively contain possible pests’ outbreaks [45, 80].
539 Moreover, about 70% of the predicted climate possibilities for the future are already al-
540 leged to be occurring on earth in some places, thus adjusting a ‘climate analogues’ method
541 spearheaded by CGIAR is contextually valuable [81, 82]. Therefore, escalating this method
542 or approach to include pests would be pivotal to understanding what species of pests
543 could be challenging for crops, and localities, cropping patterns, and henceforth inform-
544 ing appropriate changes in extension services systems [83].
I find the above difficult to understand, and here are my comments: The paragraph has been re-written more concisely, and the term “climate analogues” also clearly defined in the revised manuscript.
“Therefore” implies a conclusion based on the previous statement, a what we have here is a non-sequitor. (538) Advise was noted and incorporated in the revised manuscript.
What does this actually mean? “About 70% of predicted climate possibilities” (539) This is elaborated in the paper in parentheses besides the term possibilities.
The language is too diffused and inprecise. What does this actually mean? “are alleged to be occurring on earth in some places” (540) [Alledged?] [on earth? if not on earth, where else?] [in some place?] The sentence is re-written more concicely.
What is the ‘climate analogues’ method? “(540) defined in the paper.
Is climate analogues a method or an approach? (541/542). This is a climate monitoring methodology but based on internet applications – the term is well defined in the revised manuscript.
“Therefore” implies a conclusion based on the previous statement, a what we have here is, again, a non-sequitor. (541) Advise well noted, and effected.
563 what does "sustainable management of pests in low, " means? Is was wrongfully emitted for in, but now rectified.
The revised text is much better but certain sections are still hard to follow. I suggest that the entire document be edited by a native English language technical writer, to make the text more direct and concise, so that ideas, concepts, and conclusions are easier for the reader to follow. Also to correct misspellings, grammar and style. The entire document has been re-written with more concise wordings that ensure the fluidity of the write-up. Native English speaker has also supported reading through the revised version to ensure proper language.

Reviewer 2 Report
Authors have ignored reviewer's comments. Furthermore, this paper is better to be submitted to non-scientific journals for marketing, as it does not fit in any scientific journals.
Author Response
REVIEWER TWO
Authors have ignored reviewer's comments. We addressed all comments raised by each reviewer in the first phase, as can be seen from our point-by-point submission while we resubmitted the paper. Unfortunately, the Honourable reviewer was not particular on what comments we hadn’t addressed. Furthermore, this paper is better to be submitted to non-scientific journals for marketing, as it does not fit in any scientific journals. I differ from the reviewer on this point – since there are many disciplines of science, including social sciences. The paper assesses scientifically using a review methodology the social aspects that determine CS-IPM deployment. Even though the review is qualitative, it is still scientific as the mother papers a number of them were quantitative ones. Moreover, qualitative methods do also yield plausible qualitative scientific papers as this one.

Reviewer 3 Report
The manuscript have been improved but still it seems to me a bit too general. There are many statements which are commonly known truths. It could benefit if there are more details (for example in the form of numerical summaries, some statistics).
Author Response
REVIEWER THREE
The manuscript have been improved but still it seems to me a bit too general. We re-write the manuscript, making it sharper and more concise with now even reduced page numbers, a lot of redundant detail has been removed. There are many statements which are commonly known truths. This is the purpose of the review – that we are not generally yielding new knowledge, but gathering scattered knowledge about a particular aspect, and the more we find that the conclusions are similar across many papers (common truths) the more we confirm that such is indeed the truth. It could benefit if there are more details (for example in the form of numerical summaries, some statistics). This is basically a qualitative paper which we clearly mention in our methodology, so we deal more with statements than figures. Moreover, in sections 2.1, 2.2, and results from areas like factor 1.5 and 1.10 where statistics and figures are necessary and were found in the mother papers, these have been included. Finally, in Figure 1, and Table 1 – we give the statistics about the numbers of papers considered here, which are the only substantial stats pertinent to this paper, given the methodology used for its compilation – desk review of the findings.

Round 2
Reviewer 2 Report
This paper reports a meta-analysis which is totally unclear in materials and methods section that authors used which statistical procedures in this study?
In addition this section is so brief?
which meta-analysis papers authors used to prepare their paper? these should be referenced
Author Response
REVIEWER TWO
This paper reports a meta-analysis which is totally unclear in materials and methods section that authors used which statistical procedures in this study?
We had briefly elaborated on this method following [18, and 19], but now we expand it clearly stating it’s specific nomenclature as PRISMA. Please see the insertions in the revised manuscript with Track changes, lines 165 – 176, specifically reading as: We show our simplified identification and selection criteria of considered studies in Figure 1, following the Preferred Reporting Items for Systematic Reviews and Me-ta-Analysis (PRISMA) methodology elaborated by Moher et al. [18] and Page et al. [19]. The PRISMA methodology of meta-analysis review follows a checklist of items that we considered throughout the review to improve the needed transparency in while execut-ing such reviews. Considered items comprise all features of a plausible manuscript tar-geted for the review, for instance the title of the manuscript, its abstract, its introduction, methods used in the manuscript, results reported, discussions of the reported results, and sometimes the funding sources. The objective of the PRISMA method of me-ta-analysis reviewing is to enable a thorough review of all possible original studies that empirically and soundly analyzed the subject matter, in this case, the CS-IPM so-cio-economic aspects globally.
In addition this section is so brief?
The section is now given more flesh to elaborate what the PRISMA methodology is, what it considers and why it is preferred. Please see lines 165 – 175 in revised manuscript with Track changes
which meta-analysis papers authors used to prepare their paper? these [ referenced
These have been referenced and cited in the methods sections – two papers that technically developed the methodology [18 and 19], and two papers that recently used the methodology [16 and 17].
